# Research on social bot identification through behavioral feature analysis

**Peng Zhang**[ID][1]*, **Yinghui Du**[1], **Qilei Wang**[ID][2], **Jiyang Zhang**[1], **Ruiqing Qin**[1]

**1** Network Public Opinion Research Center of China People's Police University, Langfang, China,
**2** Research Centre for Modern Police Technology and Equipment of China People's Police University, Langfang, China

* zhangpeng@cppu.edu.cn

## Abstract

Accurately identifying social bot accounts is the key to preventing the use of artificial intelligence technology to forge social accounts, which can interfere with public opinion and thus cause public opinion crises. However, at present, relying only on manual identification of bot accounts has the challenges of low efficiency, high cost, and low accuracy, while existing research on batch identification of social bots lacks research on the system of behavioural characteristics of social bots, and thus lacks the construction of a model for the analysis of the behavioural characteristics of social bots. In this paper, we propose a diverse set of behavioural features for social robots based on the differences between the behavioural features of social robot accounts and normal users. The feature selection method based on OOB estimation is chosen for excluding redundant features in the constructed feature set; meanwhile, Random Forest, as a combined classification method, overcomes the problem of limitations of decision boundaries when classifying with a single decision tree, and has the characteristics of high accuracy, fast speed and stable performance. Through experiments, this paper applies it to the construction of social robot recognition model for detecting robot accounts in social platforms. The experiments prove that the effective indicators screened by the feature selection method based on OOB estimation can help improve the stability of the model. Specifically, the filtered features contribute about 20% more to the model accuracy and F1 score than other features. The social robot recognition model constructed based on random forest has higher accuracy and stability compared to the decision tree model and neural network model. Specifically, the accuracy rate is about 5% higher than other models, and other indicators are also better than other models. The experimental results show that the feature selection method based on OOB estimation and the random forest model show excellent performance in the experiments of social robot recognition, which can meet the requirements of the actual social robot recognition research and can be applied to the practical scenarios of robot account detection on social platforms.

**Data availability statement:** All relevant code is available via GitHub at the following URL: https://github.com/lnvadev/socialbot1. Additional data can be found in this manuscript's File Inventory and in the body of the manuscript itself.

**Funding:** This work was supported by "Humanities and Social Sciences Foundation of the Ministry of Education (No. 22YJA860012) to P.Z.", "Humanities and Social Science Research Project of Hebei Education Department (No. BJ2025339) to P.Z. funded by Science Research Project of Hebei Education Department". We are grateful to the anonymous reviewers for their constructive comments that improving the quality of this work.

## 1. Introduction

A social robot is a kind of virtual AI image running on social media platforms that simulates human social behaviour and participates in human social activities [1]. In recent years, social robots supported by AI technology have been widely used in various online social platforms for the purpose of creating topics, diverting attention, and intervening to manipulate online public opinion. Social robot technology has been applied to a variety of fields, including economics, politics, health care, etc. [2–4], for example, according to statistics, during the 2018 U.S. presidential election, social robot accounts accounted for 21.1% of all Twitter users, and the total number of tweets sent by these social robot accounts accounted for 30.6% of all election tweets, which interfered with the results of the U.S. election directly or indirectly [5]. For the economy, 71% of Twitter users involved in predicting US stock market trends were bot accounts, manipulating the stock market [6]; the average presence of bots in active Facebook accounts in 2019 was 11%, interfering with normal public opinion [4]. Driven by various factors such as political or economic interests, the number or proportion of social bots is still showing an increasing trend, while some studies have found that current intelligent technologies can support the development of social bots that interact normally with human accounts [7], social bots are becoming an important factor influencing public opinion, and social bots amplify the scale, scope, and speed of information disorders, which need to be taken seriously and governance [8].

Currently, foreign scholars have achieved some results in the field of social robot identification. Wang et al [9] used crowdsourcing method to build a multi-layer detection system, and verified the effectiveness of crowdsourcing method; Alarifi et al [10] constructed a classifier by labelling the account attributes; Dickerson et al [11] introduced a sentiment analysis model and proved that the sentiment factor is the key; Chensu et al [12] propose an embedding model based on graph attention networks; Hurtado et al [13] discover social bots manipulating public opinion through abnormal behaviour analysis; Daouadi et al [14] construct a recognition model based on deep learning to improve detection efficiency.

Domestically, Zhang Yanmei et al [15] proposed a network navy identification model based on features such as fan attention ratio; Hu Fanggang et al [16] improved the random forest algorithm to enhance the accuracy by using dynamic features; Li Yangyang et al [17] introduced generative adversarial network to achieve high efficiency detection; Wu Peiyin et al [18] showed that social robots can manipulate the public opinion; and Lu Linyan et al [19] discovered social robots based on model fusion significant features in hotspot events. Research methods mainly include crowdsourcing, machine learning and graph-based detection methods [20–22]. Among them, crowdsourcing methods have high labour cost and low efficiency; machine learning methods make use of sentiment features, but are mostly coarse-grained analysis; graph methods are limited by data collection and have limited performance.

With the development of artificial intelligence technology, social robot forgery is becoming more and more covert, and public opinion manipulation of hot events

may trigger online public opinion crises. However, the current recognition technology is still in the exploratory stage, with the problems of low efficiency and insufficient precision. In the future, it is necessary to construct a comprehensive behavioural feature system and improve the recognition accuracy in order to restrict the activities of social bots and maintain the stability of the network environment.

In this paper, we study the diversity of behavioural features of social robot accounts, put more emphasis on capturing the differences between the behavioural features of social robots and normal users for social robot identification, and construct a collection of behavioural features of social robots that can be better reflected; in the process of social robot identification, the quality of the features used in the construction of the model directly determines the accuracy of the identification model, and the behavioural feature attributes of social robots are complicated, which can easily cause the existence of the identification model. In the process of social robot recognition, the quality of the features used to construct the model directly determines the accuracy of the recognition model, and the attributes of the behavioural features of social robots are complicated, which can easily lead to the overfitting problem of the recognition model and thus reduce the accuracy of recognition, so how to select effective indicators more conducive to the recognition of social robots, and eliminate redundant features become the key to the construction of the recognition model.

At the same time, due to the large amount of data used for model training, the characteristics of noise, fuzzy and random, which also makes the selection of recognition model is particularly important. Therefore, this paper chooses a combination of classification algorithms robust to data noise, accurate and stable prediction results – Random Forest, Random Forest overcomes the problem of the limitations of the decision boundary in the classification of a single decision tree, and has the characteristics of high accuracy, speed and stable performance, based on which, this paper applies it to the social robot recognition model construction for detecting bot accounts in social platforms.

The experimental results prove that the set of indicators selected by the algorithm has obvious rationality and improves the status quo of the lack of objective criteria for indicator selection. At the same time, the social robot recognition model built based on this algorithm has a certain degree of improvement compared with other models in terms of prediction accuracy and stability, and more satisfactory results are obtained.

## 2. Related works

At present, foreign scholars have achieved certain results on social robot recognition research, for example, Wang et al [9] used crowdsourcing method to establish a multi-layer detection system for social robots, and the excellent detection results proved that crowdsourcing-based recognition methods are very effective. Alarifi et al [10] explored in depth the performance characteristics of the social robot accounts on the Twitter platform, and manually collected based on crowdsourcing method, labelled a large number of account attributes to construct an effective and practical classifier. Dickerson et al [11] argued that sentiment factors can well distinguish human accounts from bot accounts and introduced features based on sentiment analysis to build a recognition model, and the results showed that sentiment factors are the key to recognising bots, which can significantly increase the area under the ROC curve. Chensu et al [12] proposed a graph-based attention network semi-supervised graph embedding model, which builds an effective bot account recognition model by capturing account features and relationship features between accounts in a social network to construct a social graph of accounts. Hurtado et al [13] hypothesized that a group of social accounts with similar abnormal behaviours and with a highly correlated social network structure are social bot accounts, and through an in-depth analysis of a large number of accounts with similar abnormal behaviours, it was found that a large number of accounts with similar abnormal behaviours can be identified as social bot accounts. Hurtado et al [13] hypothesized that a group of social accounts with similar abnormal behaviours and a highly correlated social network structure are social robots, and through an in-depth analysis of a large number of accounts with similar abnormal behaviours, a group of social robots that are used to influence the public opinion are discovered. Daouadi et al [14] constructed a robot identification model based on deep learning algorithms, and used the content information of the account and the account's social behaviours as a dataset to train the

model, and the experimental results proved that the model can efficiently identify the robot accounts and normal user accounts.

With the development of China's Internet technology, domestic research on the identification of social robots (network water armies and abnormal accounts) has gradually emerged, for example, Zhang Yanmei et al [15] proposed six feature attributes such as fan attention ratio, average number of microblogs published, and comprehensive quality assessment on the basis of related research, and established a network water army identification model based on Bayesian model and genetic optimisation algorithm. Hu Fanggang et al [16] used the relevant features shown in the dynamic change process of user accounts (e.g., there are features such as the amount of change in the number of concerns and the rate of change of fans) and established a social robot account detection model based on the improved Random Forest algorithm, which significantly improved the accuracy of the detection model. Yangyang Li et al [17] proposed to use the discriminator in generative adversarial network for machine account detection, and only need to use the data of real accounts to train a good recognition model, and the experimental results show that the social robot recognition model based on generative adversarial network achieves excellent classification results. Wu Peiying et al [18] take the tweets related to the diplomatic boycott of the Beijing Winter Olympics event on the Twitter platform as the research object, and experimentally prove that the agenda networks of humans and robots are significantly correlated, and that social robots have the ability to manipulate public opinion. Lu Linyan et al [19] took a total of 11 hotly debated social and public events in 2019–2020 as the research object, and established a social robot identification model based on model fusion and other methods, and the results of the study showed that the social robots active on the microblogging platform showed obvious characteristics in hot social events, and their main purpose was to expand the scope of influence rather than to guide public opinion.

At present, scholars at home and abroad have done a lot of research work in identifying social robots and summed up relevant experiences, which are mainly summarised as the following methods: crowdsourcing-based identification method, machine learning-based identification method, graph-based detection method [20–22]. Crowdsourcing-based recognition methods refer to the use of human resources to complete the recognition task of social robots, i.e., using human intelligence to cope with artificial intelligence. Machine learning-based recognition methods refer to the purpose of converting user account information into machine-recognisable feature values for models to learn and train in large quantities, in order to make the models can accurately distinguish between human accounts and social robot accounts. A graph structure can characterise the relationships between social media accounts, and graph-based detection methods require the construction of a relationship graph between users for the identification of bot accounts and normal accounts. Specifically, by describing the different social association structure patterns between social robots and normal users, the social robot detection problem is transformed into a node classification problem in the graph, and then classified by graph mining algorithms or using machine learning algorithms as a way of distinguishing between normal accounts and social robot accounts.

In summary, crowdsourcing-based recognition methods have high labour cost and time consumption, and are also tough challenges for those performing the classification task, with significant limitations. Meanwhile, social bots can be generated quickly and in large quantities by relying on existing AI technology at a low cost, and they can also continuously change the performance characteristics of the accounts based on technological development to enhance their invisibility. In contrast, crowdsourcing-based identification methods are clearly not able to meet realistic social bot detection needs. Researchers have noted that the emotional factors implied in tweets are very different between normal users and social robot accounts [23], and the current machine learning-based recognition methods are all about coarse-grained emotion segmentation, such as extracting the emotional polarity of blog posts or the intensity of emotions as emotional features, and have not yet investigated the impact of fine-grained emotion segmentation on social robot detection. Social relationship is an important attribute of social network users, but due to the limitations of real-life social network platforms on data collection, graph-based detection methods often do not fully achieve the expected performance [23].

 

# 3. Analysis of behavioural patterns of social robots

## 3.1 Social activity patterns of social robots

Although artificial intelligence technology can help social robots better imitate normal users in social activities, the purpose of social robots in social activities is to intervene in manipulating online public opinion, creating topics, diverting attention, etc., which leads to the existence of robot accounts in the process of carrying out social activities with a different pattern of activities from that of normal users.

Fig 1(a) and 1(b) show the trend graphs of the number of followers of normal users and bots, respectively, with the cumulative login time of the accounts. It is found that the number of followers of normal user accounts rises with the increase of login time, while the change that occurs in the number of followers of bot accounts is not significant. The reason for this analysis is that normal users tend to have a certain amount of followers due to the influence of offline life and social behaviours, while most normal users have a more complex network of social activities, and the number of followers of an account will continue to increase along with the social activities.

Fig 2(a) and 2(b) show the distribution of the ratio of the number of followers to the number of followed for normal users and bot accounts, respectively. Overall the ratio of the number of followers to the number of followed of the bot account

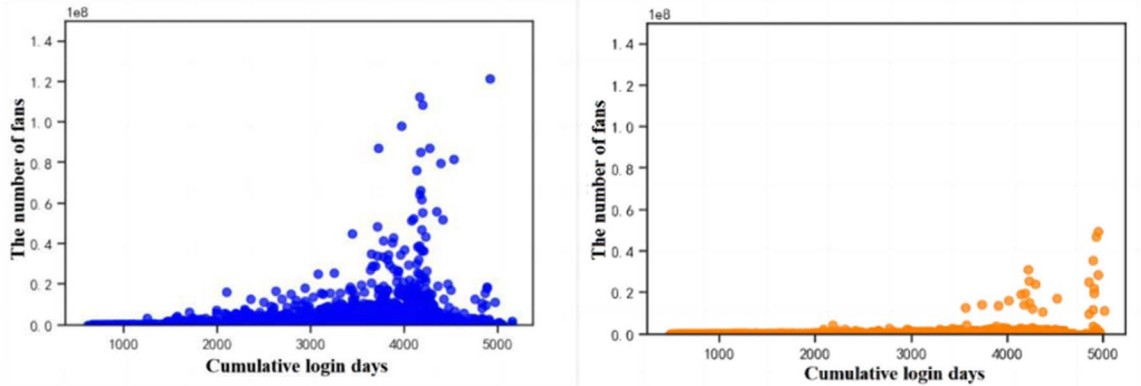

**Fig 1. Distribution of fans.** (a) Distribution of fan counts for human accounts (b) Distribution of fan counts for bot accounts.

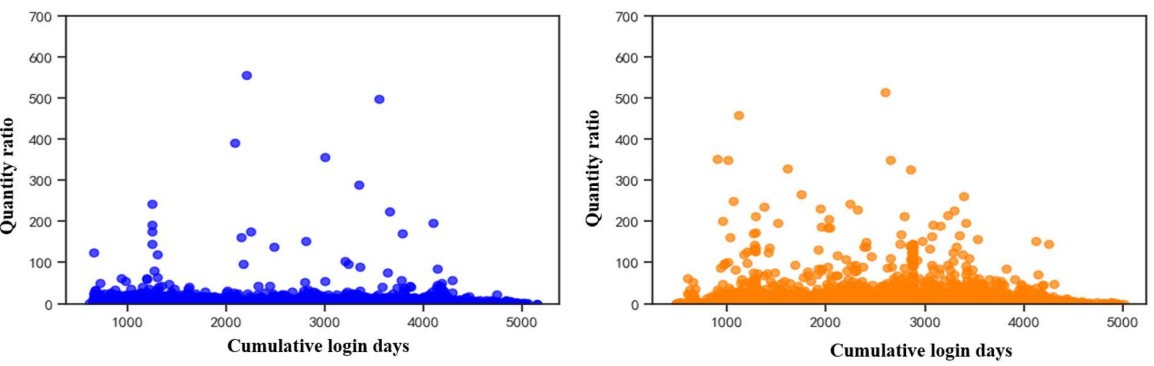

**Fig 2. Distribution of the ratio of followings to fans.** (a) Ratio of followings to fans for human accounts (b) Ratio of followings to fans for bot accounts.

is significantly higher than that of the normal user's. The reason for this analysis is that the number of followed and the number of followers of the social bot account show a strong correlation, whereas the correlation between the number of followers and the number of followed of the normal user is weaker [24], and there is a self-improvement mechanism in the social bot system, which is to increase the possibility of being followed by actively following other people. The more the number of followers, the more the number of fans will increase; while normal users have the phenomenon that the more the number of followers, the less the number of followers, which leads to the ratio of the number of followers to the number of followers of normal users is generally lower than that of the bot accounts.

Fig 3(a) and 3(b) show visual analyses of the data on normal user and bot account likes, respectively. It is found that the normal user group is significantly more active in liking activities than the social bots, and the social bot group carries out liking activities with the aim of presenting their accounts more like human users and avoiding basic bot detection, but the frequency of social activities carried out by the social bot group is lower than that of the normal user group.

Fig 4(a) and 4(b) show the distribution of the number of posts analysed for normal user accounts and bot accounts, respectively. Overall the average number of posts and the total number of posts of social robot accounts are significantly higher than those of normal user accounts. The reason for analysing this is that social robot accounts often publish or retweet tweets in large quantities for the purpose of expanding information, amplifying their position, and guiding public opinion, etc., which is fundamentally different from the purpose of normal users' tweets, which leads to the fact that the number of posts of social robot accounts will be high overall.

Fig 5(a) and 5(b) analyse the changes in the ratio of the total number of posts to the number of followers and fans for normal users and bot accounts, respectively. The results show that it is found that there are cases where the ratio of the number of posts to the number of followers and the ratio of the number of posts to the number of fans of social robots are significantly higher than the normal values, so that the probability of an account belonging to a social robot account increases when the ratio of the number of posts to the number of followers and the ratio of the number of posts to the number of fans of the account are significantly higher than the normal values.

Fig 6 shows the changes in the ratio of the number of likes to the number of concerns and the total number of posts of the analysed accounts, respectively, and the results show that the ratio of the number of likes to the number of concerns and the ratio of the number of likes to the total number of posts of social robots are found to be significantly higher than the normal value, so when the ratio of the number of likes to the number of concerns and the ratio of the number of likes

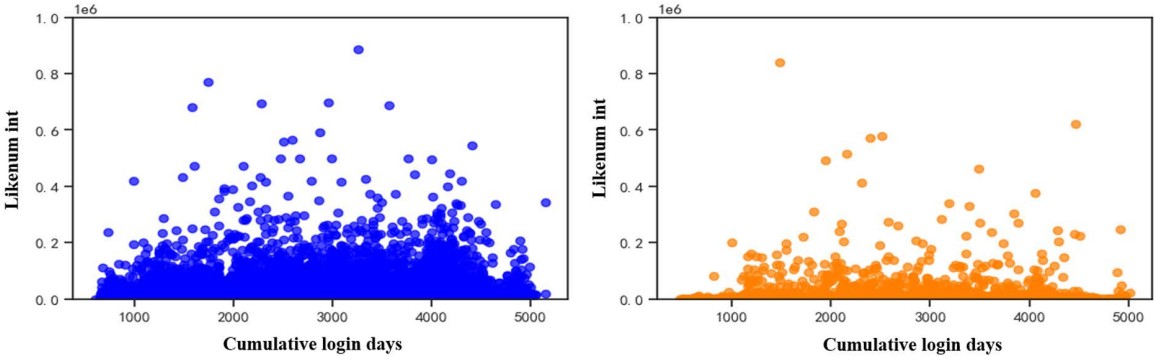

**Fig 3. Distribution of the number of likes of both accounts.** (a) Number of likes for human accounts (b) Number of likes for bot accounts.

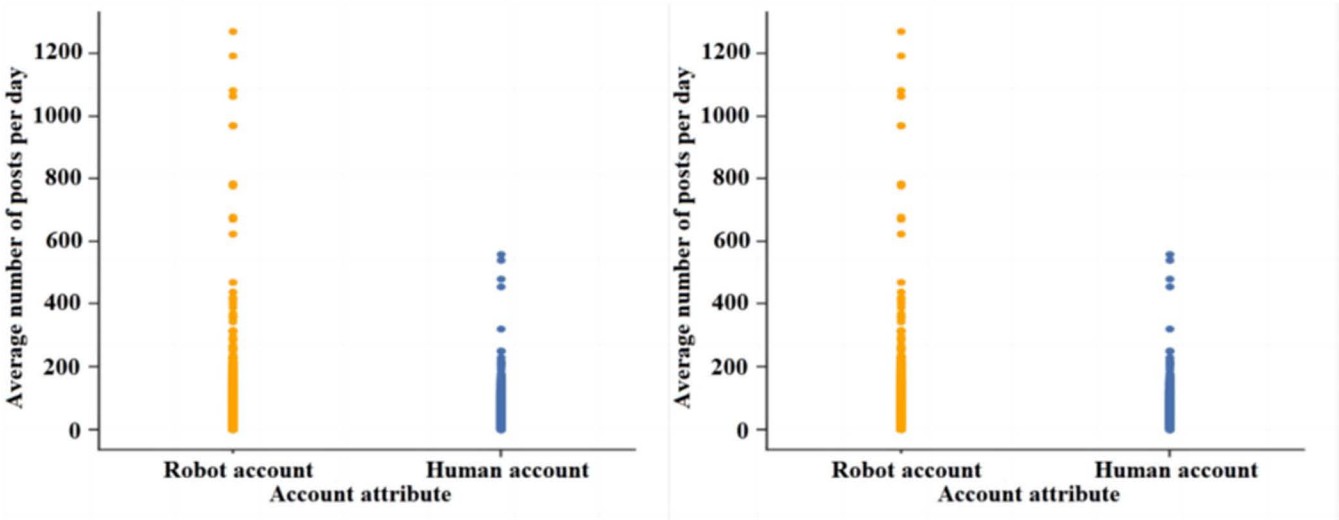

**Fig 4. Distribution of posts of both accounts.** (a) Average total posts per day (b) Cumulative total posts for accounts.

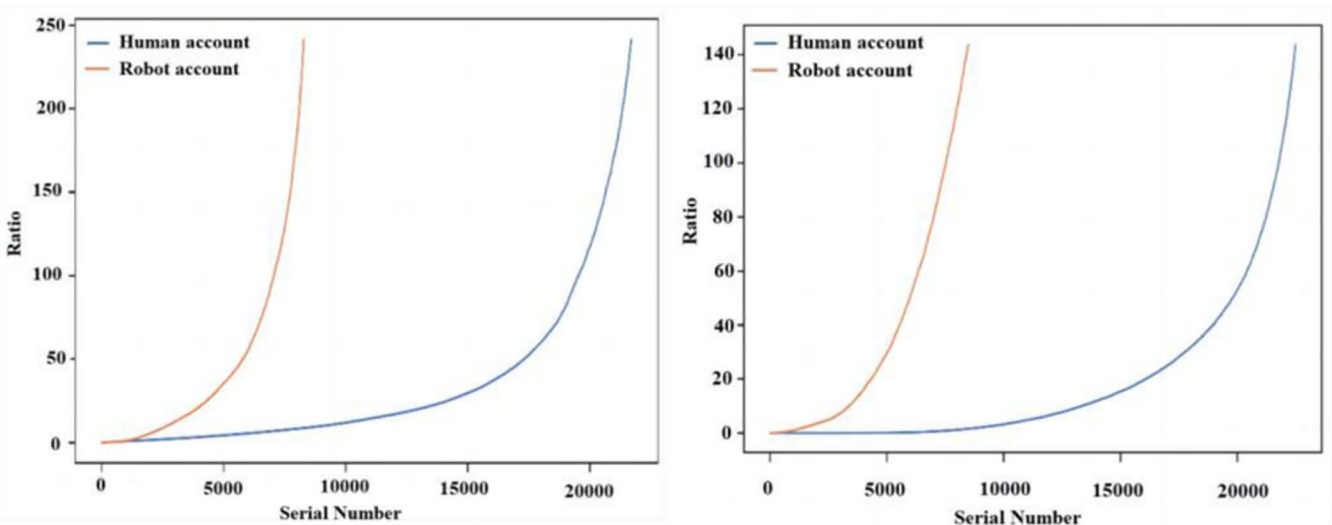

**Fig 5. Trend analysis of the ratio of total post count to number of following and number of fans.** (a) Ratio of total posts to number of following (b) Ratio of total posts to fan counts.

to the total number of posts of the account is significantly higher than the normal value, the suspicion that it belongs to the account of the social robot will beincrease.

## 3.2 Distribution pattern of account attributes of social bots

Fig 7 is a radar chart of the number of statistics on the number of social robot accounts and normal user accounts with normal or abnormal performance of basic information, respectively, including: whether the account was created on the

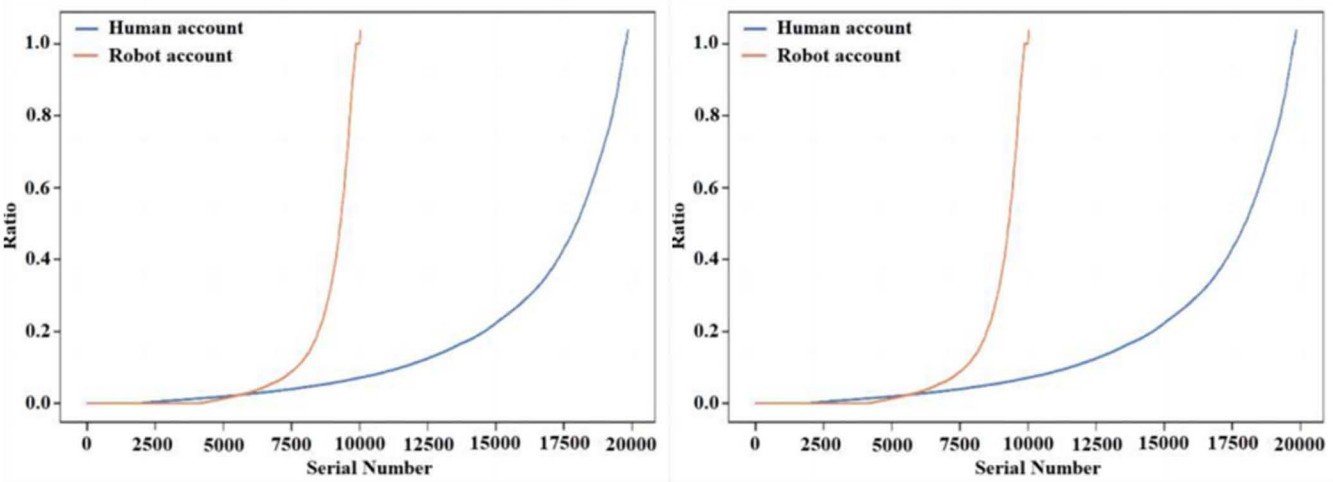

**Fig 6. Changing trends in the ratio of likes to number of followings and total post count.** (a) Ratio of likes to followings (b) Ratio of likes to total posts.

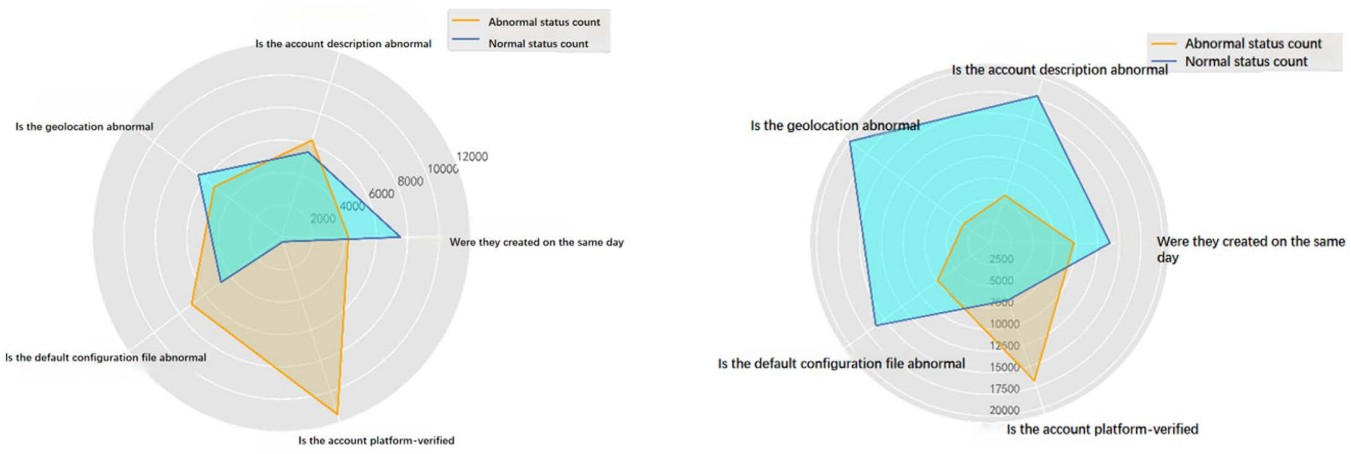

**Fig 7. Radar chart of basic account information.** (a) Social bot accounts (b) Human accounts.

same day, whether the account was authenticated by the platform, whether the account used the default profile, whether the account through geo-location was abnormal, and whether the account description was abnormal.

According to the visualisation results in Fig 7, it can be found that most normal users will be authenticated on the platform, while the vast majority of social bots' accounts will not be authenticated on the platform; the default profiles of most normal users' accounts will not show anomalies, while most of the social bots' accounts are in an anomalous state; and the vast majority of normal users' accounts' geo-locations and account descriptions are in a normal state, while The geolocation and account description of a large number of social bot accounts show abnormal status. Therefore, when a

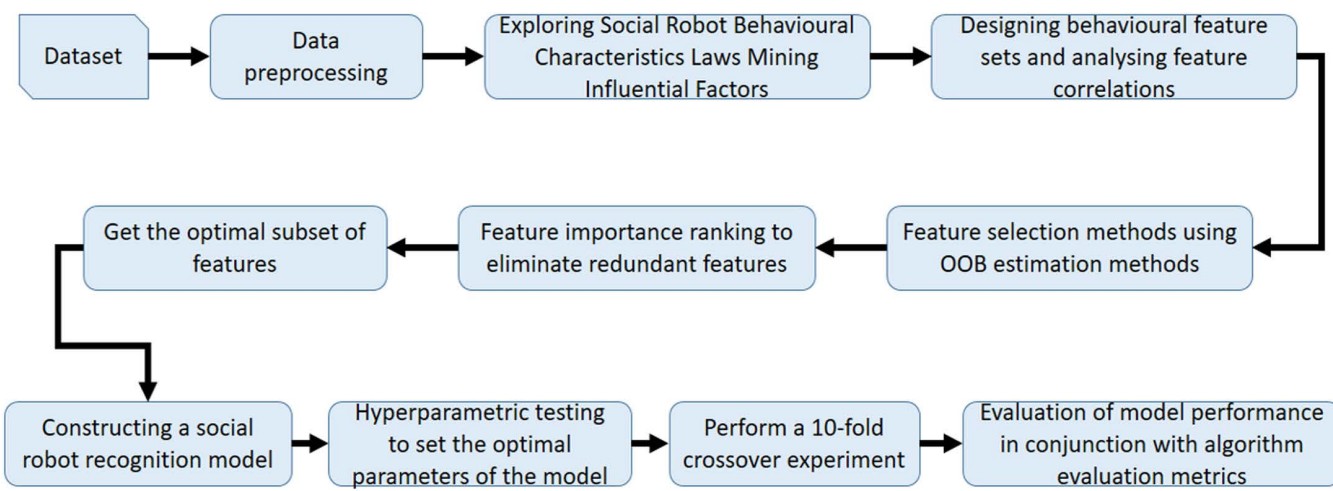

**Fig 8. General flowchart for building a social robot recognition model.**

user account has no authentication, abnormal default profile, abnormal geolocation, and abnormal account description, the suspicion that it belongs to a social bot account increases.

### 3.3 Ethical approval & informed consent

Not applicable.

This article does not contain any studies with human participants performed by any of the authors.

## 4. Construction and realisation of the model

### 4.1 Overall process

The overall process of building a social robot recognition model based on behavioural feature mining is shown in Fig 8.
There are several main steps:

Step 1, data preparation: in order to improve the information value of the dataset, data preprocessing is needed to correct outliers, eliminate redundant data that are meaningless to the study, and perform feature conversion.

Step 2, feature set design: use python tools to carry out exploratory analysis of the behavioural characteristics of the social robot account performance law, through a reasonable analysis of the research after mining from the massive data to the important influencing factors. According to the analysis results, the preliminary design of the feature set that can reasonably reflect the difference between normal user accounts and social robot accounts.

Step 3, feature selection: OOB estimation method is chosen to evaluate the importance of features, the importance of each feature is calculated and sorted in descending order, the candidate feature set is set and verified to exclude the features with lower degree of importance, and the remaining features with high importance are recombined to form an optimal feature set to participate in the model construction.

Step 4, model construction and comprehensive evaluation: using the optimal feature set obtained in step 3, the social robot recognition model is constructed by the random forest algorithm, hyperparameter tuning is carried out to obtain the optimal parameter settings of the model, and finally, the model is comprehensively evaluated by the 10-fold cross-validation experiments and combined with the relevant algorithm evaluation indexes.

## 4.2 Importance of OOB calculation characteristics

The specific steps for calculating feature importance using OOB estimation method in feature selection are as follows.

Random forests use Bagging method to make an OOB error estimate available for each decision tree, and averaging the OOB error estimates of all the decision trees gives the generalisation error of the random forest. The feature importance measure based on the classification accuracy of out-of-bag data is defined as the average reduction of the classification correctness of the features of out-of-bag data after random perturbation and the classification accuracy before the change, and the remaining features are used to observe the change of the OOB estimation of the Random Forest, and if the classification accuracy of the feature is significantly reduced after the feature has been perturbed, it indicates that the feature is strongly correlated with the corresponding target.

Assuming that the self-sampling collects samples b = 1,2,...,B, with B denoting the number of training samples, feature $X_j$ The variable importance measure based on classification accuracy is calculated as follows [25]:

In step 1, set b = 1, create a decision tree $T_b$ on the training sample, and label the out-of-bag data as $L_b^{oob}$. Use the decision tree $T_b$ to classify the out-of-bag data $L_b^{oob}$, count the number of correct classifications and record them as $R_b^{oob}$.

Step 2: Perturb the values of the features $X_j$ (j = 1,2,...,N) in $L_b^{oob}$, the perturbed dataset is labelled as $L_{bj}^{oob}$, use the decision tree to $T_b$ classify the out-of-bag data $L_b^{oob}$, and count the number of correct classifications as $R_{bj}^{oob}$.

Step 3, for b = 2,3,........,B, sequentially extract the sample set in b to be divided into training samples and out-of-bag data, and repeat step 1, step 2.

Step 4, the variable importance measure $\overline{D_j}$ of the features $X_j$ is computed by the following equation:

$$\overline{D_j} = \frac{1}{B} \sum_{i=1}^{B} \left( R_b^{oob} - R_{bj}^{oob} \right)$$

(1)

## 4.3 Random forest process

Random Forest [26] is a classifier algorithm integrated by a large number of decision trees, Random Forest has two important randomnesses, one is the randomness of training set extraction, i.e., the training and prediction of each decision tree relies on independently extracted sample data, and the other is the randomness of node candidate segmentation feature set, i.e., the features of each node in a single decision tree are randomly extracted from the feature set. Randomly extracted, the final classification result is determined by the number of votes of all tree classifiers, the random forest generation process is as follows [25]:

(1) Apply the self-help sampling method (Bootstrap) to have put back to randomly draw K new self-help sample sets, respectively, to construct K classification regression trees, each time the samples that are not drawn form K out-of-bag data sets (Out-of-bag, OOB), each time randomly sampling about 36.8% of the data are not drawn, these out-of-bag data as a test set for calculating the model's generalisederror rate or calculate feature importance, and the experiments prove that the out-of-bag data error estimation is unbiased estimation [26].

(2) Select candidate split features, assuming that there are M features, m (m ≤ M) features are randomly selected at each node of each tree, calculate the amount of information of each feature and select the best feature for node splitting, and carry out the construction of each branch of the decision tree by continuously repeating the above steps until the entire set of features is traversed.

(3) In order to eliminate the tree bias, to maximise the growth of each tree without pruning.

(4) Classification is performed using the generated multiple classification trees to form a random forest classifier, each tree returns a classification result, and the class with the most votes from the tree classifier is the final classification result.

# 5. Experiments and results

## 5.1 Introduction to the data set

The data used in this paper comes from the Social Bot Database (Bot Repository), the original dataset records in detail the specific information of user accounts that exist on social platforms, the dataset contains 37436 data about user account related information, including 12425 bot accounts and 25013 normal user accounts; the account related features in the dataset are described in detail See Table 1, for example, there are account creation time, account setup status, number of account followers, number of fans, and account specific description.

Due to the presence of redundant features in the dataset that hold no relevance for the identification research, such as "Id" (Account ID), "Profile background image url", "Profile image url", and "Screen name", these attributes have been sensibly removed. Similarly, the "location" and "lang" columns, which exhibit a significant number of missing values and lack clear relevance in distinguishing between social bots and human accounts, have also been excluded from further analysis. Moreover, during the extensive data preprocessing phase, it was observed that certain user account entries contained missing values. In light of the inability to reliably impute these missing values, a decision was made to exclude the affected accounts from the dataset. This meticulous curation process has resulted in a refined dataset comprising 35,492 informative data points. Among these entries, 11,798 accounts were classified as bots, while 23,694 were deemed as human accounts. Recognizing the class imbalance within the dataset, an intentional effort was made to address this concern by leveraging SMOTE oversampling techniques specifically on the bot account data. Consequently, this strategic oversampling led to a balanced distribution with a total of 21,324 bot account entries, aligning more closely with the count of human accounts. This meticulous utilization of social media user account data has empowered a deep exploration into the behavioral patterns exhibited by social bots. By methodically discerning and contrasting the unique behavioral characteristics of social bots against those of human users, a comprehensive and strategic set of behavior traits specific to social bots can be delineated. This refined understanding serves to significantly enhance the accuracy and efficacy of social bot detection models.

## 5.2 Feature set design and feature selection

The increase of features will enhance the complexity of the model and lead to overfitting phenomenon. Establishing a reasonable and effective collection of social robot behavioural features can improve the machine learning training speed,

**Table 1. List of dataset features.**

| Variable Name | Variable Meaning | Variable Type | Variable Name | Variable Meaning | Variable Type |
|---|---|---|---|---|---|
| Created_at | Account creation time | Numerical | location | locate | Non-numerical |
| Default profile | Whether to use the default settings | Non-numerical | Profile background image url | Use of background image sources | Non-numerical |
| Default profile image | Whether to use the default avatar | Non-numerical | Profile image url | Use avatar image source | Non-numerical |
| Description | Account Description | Non-numerical | Screen name | Account Name | Non-numerical |
| Favourites count | Number of Favourite Files | Numerical | Statuses count | Total number of posts on the account | Numerical |
| Followers count | Number of fans | Numerical | Verified | Whether or not they are certified | Non-numerical |
| Friends count | Number of mutual friends | Numerical | Average tweets perday | Average number of articles per day | Numerical |
| Geo enabled | Whether to enable geolocation | Non-numerical | Account age days | Accumulated days logged into the account | Numerical |
| Id | Account id | Numerical | Account type | Is it a bot account | Non-numerical |
| lang | longitudes | Non-numerical | | | |

reduce the influence of noise generated by irrelevant information and improve the model stability. Combined with the results of the analysis of social robot behavioural features in section 3, to reasonably build the social robot behavioural feature set, this paper introduces a total of 21 feature attributes in two categories, including social activity features and basic account information features, the feature set is shown in Table 2.

After numerical processing of the non-numerical features in Table 2, combined with the feature selection method based on OOB estimation introduced in Section 4, the average classification accuracy reduction of each feature after being perturbed is calculated separately, and the larger the average classification accuracy reduction value is, the higher the degree of importance of the feature is proved to be. In order to verify the reasonableness of the OOB estimation method, comparison is made by calculating the average impurity reduction value of the Gini index, the larger the average impurity reduction value of the Gini index, the higher the degree of importance of the features, the results of the feature selection based on the 2 methods are shown in Fig 9.

According to the results of feature importance ranking based on the OOB estimation method in Fig 9, the unique features mined in this paper contribute significantly to the model accuracy, for example, the ratio of the number of likes to the total number of posts (NO.110) contributes 13.8%, the ratio of the number of followers to the cumulative number of logged-in days (NO.111) contributes 7.2%, and the total number of posts to the number of followersratio (NO.108), the ratio of the number of followers to the number of cumulative login days (NO.113), and the ratio of the number of likes to the number of cumulative login days (NO.112) all contribute more than 5%, the ratio of the number of followers to the number of followed (NO.105), the ratio of the total number of posts to the number of followers (NO.107) contribute more than 4%, and the ratio of the number of likes to the number of followers (NO.109) has a contribution of 3%. The results

**Table 2. Feature list.**

| Feature Type | NO. | Feature Name | Feature Type | Feature Type | NO. | Feature Name | Feature Type |
|---|---|---|---|---|---|---|---|
| Social Activity | 101 | Number of concerns | Numerical | Basic Account Information | 114 | Whether the account was created on the same day | Non-numerical |
| | 102 | Fan number | Numerical | | 115 | Is the account description abnormal | Non-numerical |
| | 103 | Number of likes (on a website) | Numerical | | 116 | Is the geolocation anomaly | Non-numerical |
| | 104 | Average number of articles per day | Numerical | | 117 | Whether the default configuration file is abnormal | Non-numerical |
| | 105 | Ratio of number of followers to number of people followed | Numerical | | 118 | Is the default image profile abnormal | Non-numerical |
| | 106 | Total number of user posts | Numerical | | 119 | Accumulated days logged into the account | Numerical |
| | 107 | Ratio of total number of communications to number of concerns | Numerical | | 120 | Whether the account is authenticated by the platform | Non-numerical |
| | 108 | Ratio of total number of posts to number of followers | Numerical | | 121 | Whether the account is geo-location enabled | Non-numerical |
| | 109 | Ratio of Likes to Follows | Numerical | | | | |
| | 110 | Ratio of number of likes to total number of postings | Numerical | | | | |
| | 111 | Ratio of number of fans to cumulative login days | Numerical | | | | |
| | 112 | Ratio of number of likes to cumulative days logged in | Numerical | | | | |
| | 113 | Ratio of number of followers to cumulative days logged in | Numerical | | | | |

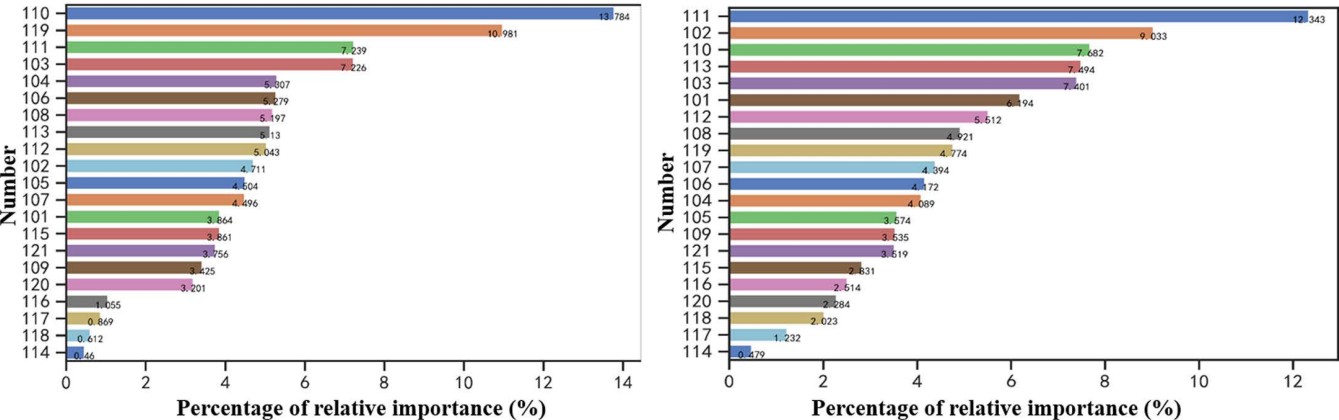

**Fig 9. Feature importance metrics.** (a) Average decrease in out-of-bag error accuracy (b) Average decrease in Gini coefficient impurity.

show that the behavioural features of social bots mined by the analysis in this paper can effectively improve the accuracy of the recognition model.

In the feature importance ranking based on the OOB estimation method, the features ranked as the top 10 features are: the ratio of the number of likes to the total number of posts (NO.110), the cumulative number of login days of the account (NO.119), the ratio of the number of followers to the cumulative number of login days (NO.111), the number of likes (NO.103), the average number of posts per day (NO.104), the total number of user posts (NO.106), ratio of total number of posts to number of followers (NO.108), ratio of number of followers to cumulative days of logging in (NO.113), ratio of number of likes to cumulative days of logging in (NO.112), and number of followers (NO.102), and comparing with the feature selection method based on the Gini index, the list of the top 10 ranked features contains 8 of the same features, and the last 5 rankedfeatures are also the same, which shows that the feature importance assessed based on the OOB estimation method is reasonable. Finally, based on the results of the OOB estimation of feature importance, the features ranked in the last 4 with low contribution are eliminated, including whether the geolocation is abnormal (NO.116), whether the default profile is abnormal (NO.117), whether the default image profile is abnormal (NO.118), whether the account was created on the same day (NO.114), and a total of the rest of the 17 features are selected for the construction of thesocial bot recognition model.

## 5.3 Parameter tuning

In this paper, Accuracy, Precision, Recall and F1-Score are chosen as the evaluation indexes of the model. Among them, Accuracy is the ratio of the number of correctly classified samples to the total number of samples, which can be used to evaluate the overall recognition rate of the model; Precision is the ratio of the number of correctly recognised samples to the total number of samples recognised, which can be used to evaluate the precision of the model; Recall is the ratio of the number of correctly recognised samples to the number of samples that should have been recognised, which can be used to evaluate the completeness of the model's recognition; F1-Score is an index of the accuracy of classification models, which can be viewed as a measure of the accuracy of classification models. The F1 score is a measure of the precision of the classification model, which can be regarded as the reconciled average of the precision rate and the recall rate, and the higher the F1 score is, the better the performance of the model is proved. The formulae are as follows:

$$Accuracy = \frac{TP + TN}{TP + TN + FP + FN}$$

(2)

$$Precision = \frac{TP}{TP + FP} \tag{3}$$

$$Recall = \frac{TP}{TP + FN} \tag{4}$$

$$F1 = \frac{2 \times Precision \times Recall}{Precision + Recall} \tag{5}$$

where TP (True Positive) denotes the number of true categories that are positive and predicted categories that are positive, FP (False Positive) denotes the number of true categories that are negative and predicted categories that are positive, FN (False Negative) denotes the number of true categories that are positive and predicted categories that are negative, and TN (TrueFN(False Negative) denotes the number of positive cases with true category and negative cases with predicted category, and TN(True Negative) denotes the number of negative cases with true category and predicted category. In this paper, we classify the account attribute "human account" as a positive case and the account attribute "social robot account" as a negative case.

According to Section 5.2, we filter the feature set with higher importance, exclude the feature columns that do not belong to this set in the original data set, and use the remaining data as the data set for constructing the social robot recognition model, respectively use the random forest, neural network, and decision tree to establish the recognition model, and adjust the parameters of each model using the method of controlling variables to determine the optimal parameters of the model, and the results of the parameter tuning are shown in Table 3 model, the final parameters are: the maximum number of features used in a single decision tree is set to 10, and the number of trees in the forest (i.e., the number of base evaluators) is set to 400; for the neural network model, the final parameters are: the number of neurons in the hidden layer (the number of nodes) is set to 50, and the initial learning rate for controlling the magnitude of each weighting parameter update is set to 0.01, and the parameters used in the optimisation process are set to 0.01, which is the optimum number of neurons in the hidden layer. The maximum number of iterations allowed was set to 200; for the decision tree model, the maximum number of feature lookups was set to 7, and the maximum depth of tree branches was set to 15.

**Table 3. Results of tuning the parameters of each model.**

| random forest | | | neural network | | | | decision tree | | |
|---|---|---|---|---|---|---|---|---|---|
| characteristic number | tree | accuracy | Number of nodes | learning rate | iteration number | accuracy | characteristic number | depth | accuracy |
| 8 | 50 | 0.8983 | 40 | 0.001 | 100 | 0.8225 | 5 | 5 | 0.8159 |
| 9 | 50 | 0.8985 | **50** | **0.001** | **100** | **0.8456** | 6 | 5 | 0.8130 |
| **10** | **50** | **0.9001** | 60 | 0.001 | 100 | 0.8394 | **7** | **5** | **0.8292** |
| 11 | 50 | 0.8976 | 70 | 0.001 | 100 | 0.8398 | 8 | 5 | 0.8221 |
| 12 | 50 | 0.8938 | **50** | **0.01** | **100** | **0.8556** | 9 | 5 | 0.8156 |
| 10 | 200 | 0.9003 | 50 | 0.05 | 100 | 0.8465 | 7 | 10 | 0.8641 |
| 10 | 300 | 0.8998 | 50 | 0.1 | 100 | 0.8043 | **7** | **15** | **0.8652** |
| **10** | **400** | **0.9012** | 50 | 0.5 | 100 | 0.8176 | 7 | 20 | 0.8538 |
| 10 | 500 | 0.8998 | 50 | 0.01 | 150 | 0.8503 | 7 | 25 | 0.8523 |
| | | | **50** | **0.01** | **200** | **0.8565** | | | |
| | | | 50 | 0.01 | 300 | 0.8565 | | | |

## 5.4 Analysis of results

According to the screened features and regulated optimal parameters, random forest, neural network and decision tree models are constructed respectively, 10-fold cross-validation experiments are conducted to verify the accuracy of the models and the evaluation indexes of each model are calculated. In order to judge the effectiveness of each model in the experiment, the benchmark model is added as a standard, and it is generally believed that only when the accuracy of the model is higher than that of the benchmark model can it be judged as a model with good performance, and the accuracy of the benchmark model refers to the maximum rate of classification correctness that can be achieved when the classification prediction experiments are conducted without the use of any indicators. From the results of the prediction experiments in Table 4, we can see that the accuracy rate of the benchmark model is about 52%, and the accuracy rates of the experimentally constructed models are much higher than that of the benchmark model, which proves that the models constructed in this paper have good performance. At the same time, the F1 scores of all three models reach more than 80%, which proves the reasonableness of the social robot feature set designed in this paper. Comparing the accuracy rate of each model, the random forest model has the highest average accuracy rate of 0.8991, followed by the decision tree model with an average accuracy rate of 0.8540, and the neural network model has the lowest average accuracy rate of 0.8473. This indicates that the random forest model has better accuracy in overall sample prediction.

Comparing the precision rate of each model, the random forest model has the highest average precision rate of 0.8981, while the decision tree model has an average precision rate of 0.8538 and the neural network model has an average precision rate of 0.8472, which are lower than the random forest model, suggesting that the random forest model has a more accurate discrimination of social bot accounts.

Comparing the recall rate of each model, the random forest model has the highest average recall rate of 0.8993, while the average recall rate of the decision tree model is 0.8532 and the average recall rate of the neural network model is 0.8461, which are lower than that of the random forest model, indicating that the random forest model carries out a greater probability of identifying the social robot accounts, and is able to more accurately identify thesocial robot accounts with abnormal behaviour.

Comparing the F1 scores of the models, the random forest model also has the highest F1 score of 0.8987, which is higher than the decision tree model's F1 score of 0.8535 and the neural network's F1 score of 0.8466. The F1 score is the sum of the precision rate and the recall rate, and a higher F1 score indicates that the prediction model is more accurate.

Table 4. Results of model evaluation indicators.

| test Number of times | baseline model | random forest | | | neural network | | | decision tree | | |
|---|---|---|---|---|---|---|---|---|---|---|
| | accuracy | accuracy | accuracy | recall rate | accuracy | accuracy | recall rate | accuracy | accuracy | recall rate |
| 1 | 0.5311 | 0.8943 | 0.8931 | 0.8945 | 0.8503 | 0.8506 | 0.8486 | 0.8552 | 0.8549 | 0.8541 |
| 2 | 0.5302 | 0.8994 | 0.8985 | 0.8997 | 0.8485 | 0.8479 | 0.8484 | 0.8556 | 0.8558 | 0.8542 |
| 3 | 0.5162 | 0.8965 | 0.8955 | 0.8967 | 0.8485 | 0.8493 | 0.8476 | 0.8607 | 0.8613 | 0.8600 |
| 4 | 0.5113 | 0.9036 | 0.9023 | 0.9038 | 0.8394 | 0.8394 | 0.8392 | 0.8478 | 0.8480 | 0.8476 |
| 5 | 0.5333 | 0.9038 | 0.9028 | 0.9043 | 0.8503 | 0.8497 | 0.8511 | 0.8550 | 0.8546 | 0.8538 |
| 6 | 0.5233 | 0.9016 | 0.9004 | 0.9019 | 0.8552 | 0.8564 | 0.8536 | 0.8634 | 0.8634 | 0.8627 |
| 7 | 0.5249 | 0.8923 | 0.8917 | 0.8923 | 0.8461 | 0.8412 | 0.8420 | 0.8461 | 0.8459 | 0.8453 |
| 8 | 0.5271 | 0.9020 | 0.9014 | 0.9021 | 0.8418 | 0.8434 | 0.8396 | 0.8492 | 0.8487 | 0.8491 |
| 9 | 0.5365 | 0.9016 | 0.9006 | 0.9022 | 0.8496 | 0.8486 | 0.8494 | 0.8580 | 0.8572 | 0.8575 |
| 10 | 0.5292 | 0.8960 | 0.8950 | 0.8959 | 0.8438 | 0.8453 | 0.8415 | 0.8487 | 0.8484 | 0.8477 |
| average value | 0.5263 | **0.8991** | **0.8981** | **0.8993** | 0.8473 | 0.8472 | 0.8461 | 0.8540 | 0.8538 | 0.8532 |
| (statistics) standard deviation | 0.0073 | 0.0041 | 0.0040 | 0.0042 | 0.0047 | 0.0050 | 0.0051 | 0.0058 | 0.0059 | 0.0058 |
| F1 fraction | | | **0.8987** | | | 0.8466 | | | 0.8535 | |

Comparing the stability of the models, the random forest model with an overall accuracy standard deviation of 0.0041 is the most stable one with the least volatility in the prediction results, while the decision tree model with an overall accuracy standard deviation of 0.0058 and the neural network model with an overall accuracy standard deviation of 0.0047 have higher instability than the random forest model.

Combining all the results obtained, the random forest model performs better results in the recognition experiment of social bot accounts.

Since the quality of the features used to construct a social robot recognition model directly determines the accuracy of the recognition model, this paper uses a feature selection method based on OOB estimation to screen out effective indicators and eliminate redundant features. In order to verify the reliability of the screening results, a variety of models were constructed for feature importance comparison experiments.

As shown in Fig 10, the random forest, neural network, and decision tree models use the eliminated redundant features to construct recognition models in the accuracy rate are only about 70%, and the performance in the F1 scores are only about 70% of the score; and using the random forest-based feature selection method selected feature set for modelling, the model's accuracy rate, F1 scores have reached more than 85%, of which the random forest model's accuracy rateand F1 score are the highest, which are about 90%. The results show that using the feature selection method based on OOB estimation can effectively eliminate redundant features and play an important role in constructing a reasonable set of social robot behavioural features.

According to the results of the analysis of the behavioural characteristics law of social robots, this paper adds some new features to be incorporated into the recognition model, among which the number of concerns, the number of followers, the number of likes, the average number of posts per day, the total number of posts, whether the account is created on the same day, whether the geo-location is abnormal, whether the default image profile is abnormal, whether the status of the default profile is abnormal, whether the account cumulative days of logging in, whether the account is enabled for geopositioning, and whether the account is authenticated by the platform belong to the features of the original dataset, and the rest of the features are newly added in this paper. In order to verify the contribution of the newly added features to the recognition model, this paper constructed a variety of models for comparison experiments, and the results are shown in Fig 11.

As shown in Fig 11, random forest, neural network, and decision tree models were constructed for feature contribution comparison experiments. For the random forest model, only using the new features in this paper for recognition

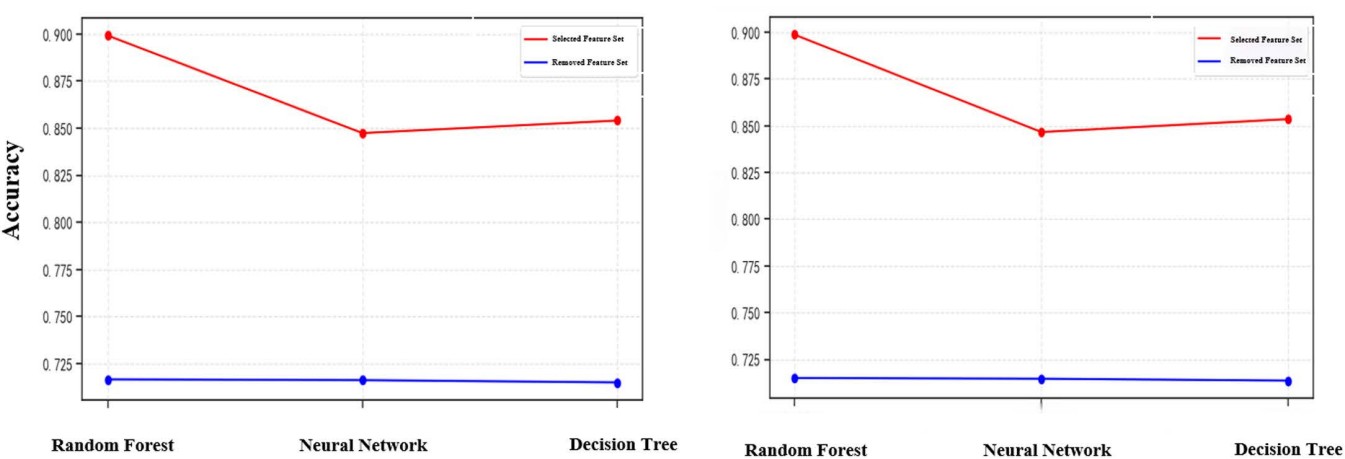

**Fig 10. Comparison of feature contribution.** (a) Comparison chart of accuracy (b) Comparison chart of F1 scores.

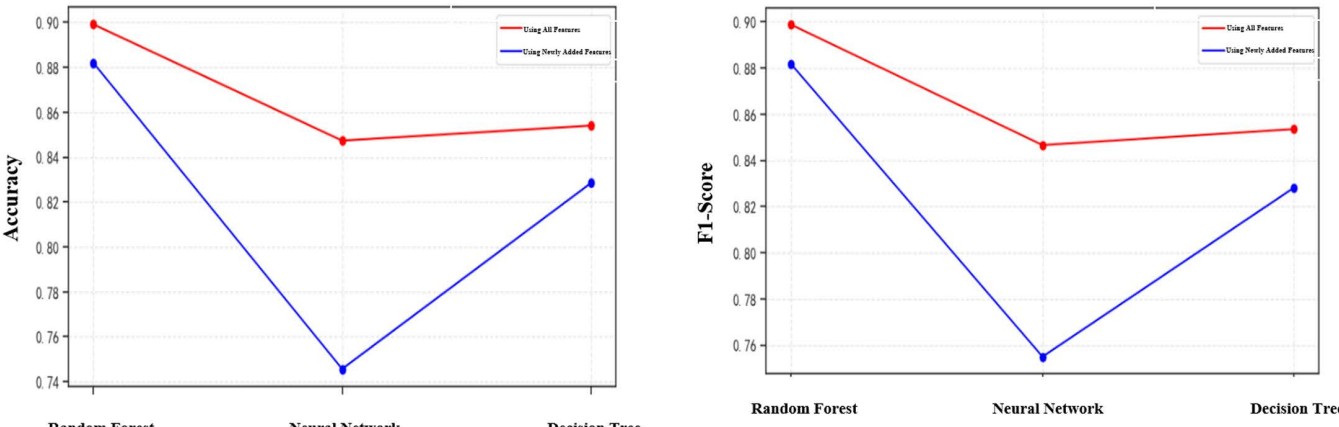

**Fig 11. Comparison of feature contribution.** (a) Comparison of accuracy (b) Comparison of F1 scores.

experiments in the accuracy and F1 score reached 88%, close enough to the results obtained by using all the features; for the decision tree model, only using the new features in this paper for the experiments in the accuracy, F1 score are about 82%, which is only about 4 percentage points lower than that of the experiments using all the features; for the neural network model, onlyFor the neural network model, the accuracy and F1 score of the experiments using only the new features in this paper are 74% and 75%, which are about 12 percentage points and 11 percentage points lower than the experiments using all the features, respectively. The experimental results show that the new features added in this paper have a high contribution to the recognition model, and the new features play an important role in helping to improve the accuracy of the model.

Finally, Random Forest, Neural Network, and Decision Tree models were constructed to perform recognition experiments on all user accounts, and the model prediction error results are shown in Fig 12.

According to the visualisation results in Fig 12, the overall prediction error value of the random forest model is the lowest in the social robot account recognition experiments, indicating that the random forest has higher recognition accuracy and stability; followed by the decision tree model, which has a better overall prediction performance, but the prediction error of some accounts is large, resulting in a decrease in recognition accuracy; the overall prediction performance of the neural network is poorer, with a higher overall error curve in the recognitionThe overall prediction performance of the neural network is poorer, and the prediction error values of more accounts in the experiment are too high, resulting in a high overall error curve, which indicates that the overall recognition accuracy and stability of the neural network model is poorer. On the whole, the random forest model is more suitable for this kind of social robot recognition research.

## 6. Discussion

### 6.1 Analysis of limitations and potential failures

The model limitations are mainly reflected in the following aspects. First, there are some limitations in feature selection and the generalisation ability of the model. The model highly relies on the set of features screened based on OOB estimation, which means that once the behavioural characteristics of social robots change, such as adjusting the frequency of posting or liking behaviour, the existing features may lose their validity. In addition, although the feature selection eliminates some redundant features, more features are retained, which may lead to overfitting of the model when the sample size is insufficient.

Second, dataset bias and representativeness issues limit model performance. The data source is mainly focused on the Bot Repository dataset, which is biased towards specific social platforms and is difficult to cover the behavioural

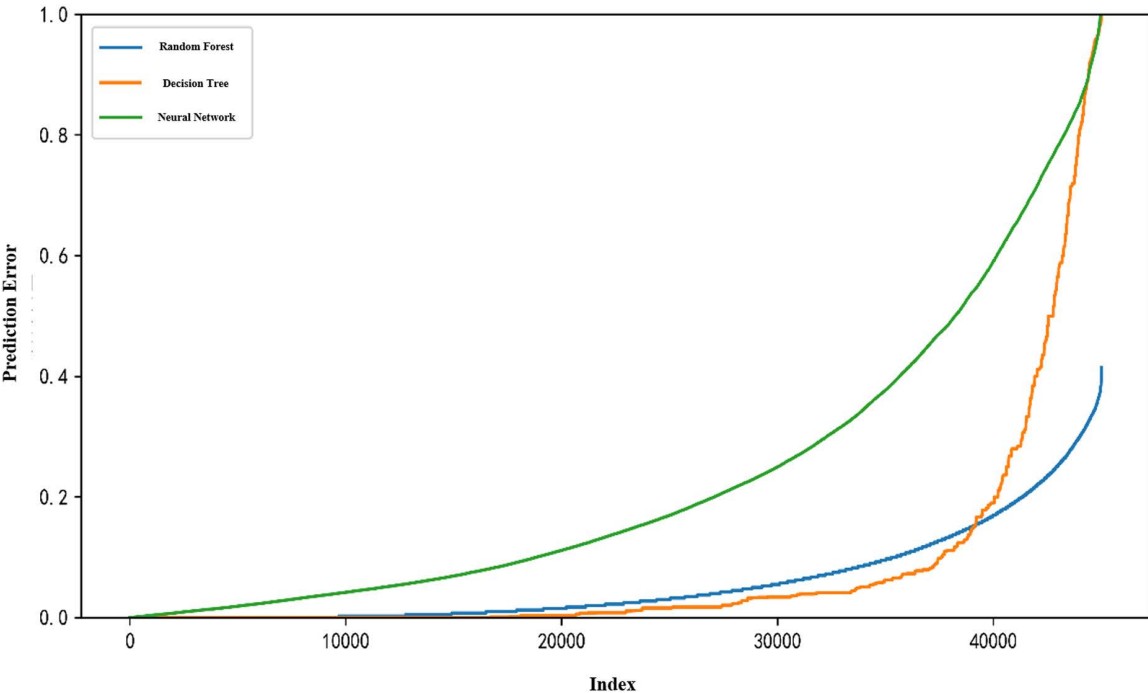

**Fig 12. Model prediction error plot.**

features of social bots on other platforms. Meanwhile, the data balance processing uses the SMOTE method for oversampling, which may introduce artificial samples, thus preventing the model from accurately reflecting account behavioural characteristics in real scenarios.

Third, there are some limitations in algorithm selection. Although the random forest model performs well in terms of robustness and stability, it is less efficient in processing large-scale and high-dimensional data, and the time cost of training and prediction is higher. In addition, the model is mainly based on behavioural features and does not fully consider the emotional tendency and semantic information in the content of social bot blog posts, which may lead to the omission of judging bots with strong camouflage.

The limitations of application scenarios are also of concern. Social bots can dynamically adjust their behaviours, but the existing model lacks the ability to adapt in real time, making it difficult to cope with new attack strategies. In addition, due to the differences in the behavioural characteristics of bots on different social platforms, the model may experience a significant performance degradation when applied across platforms.

Potential failure scenarios include highly stealthy bots that are difficult to identify. When bots mimic normal user behaviours more realistically, such as emotional speech or randomised likes, the model may have difficulty in distinguishing them effectively. Meanwhile, in highly noisy data scenarios, the model is prone to misidentify normal users as bots, leading to higher false alarm rates. If specific key features (e.g., account authentication status) are not available in real-world scenarios, the model performance will also be directly affected.

### 6.2 Improvements and prospects

Improvement directions include strengthening the sentiment analysis capability, incorporating sentiment classification and fine-grained semantic features into the model, and enhancing the recognition of complex behaviours. Introducing adaptive

learning mechanism to update model weights and features in real time to adapt to dynamic behavioural changes. Further collect multi-platform data to improve the generality and robustness of the model. Optimise the algorithm architecture by exploring efficient deep learning methods (e.g., graph neural network) to enhance the recognition effect under complex scenes.

Social robots cannot fully simulate human emotions, and the emotional tendency reflected in the social activities of actual online platforms is "unidirectional", so the features of direct emotional expression, unidirectional emotional output, and lack of emotional feedback can be used for social robot recognition. Since the modelling data in this paper lacks the information of blog posts and comments of social robots, it is impossible to explore the influence of such factors on improving the accuracy of the recognition model. Therefore, the next step is to mine the features that social robots exhibit in terms of emotional behaviour and incorporate them into the construction of the social robot recognition model to further improve the effectiveness and practical value of the recognition model.

## 7. Conclusion

This paper analyses user account information using social platforms and finds that there are significant differences between social robot accounts and normal users. Analysis of the social behaviour of social robots reveals that ① the number of fans of normal user accounts rises with the increase of login time, while the change in the number of fans of robot accounts is not obvious; ② the ratio between the number of followers and the number of followed of social robot accounts is significantly higher than that of normal users, and the number of followed and the number of followers of social robot accounts show a strong correlation, while the correlation between the number of followers and the number of followed of normal users is weak; ③ the liking activity of normal user groups is significantly more active than that of social robots, and social robot groups carry out social activities. The average number of posts and the total number of posts of social robot accounts are significantly higher than those of normal users, and the ratio of the total number of posts to the number of followers and the ratio of the number of posts to the number of fans of social robots are significantly higher than those of normal users; ⑤ the ratio of the number of posts to the number of followers of social robots is significantly higher than that of normal users; ⑤ the ratio of the number of followers to the number of followers of social robot accounts is significantly higher than that of normal users; ⑤ the number of posts to the number of followers of social robot accounts is significantly higher than that of normal users.users; ⑤ the ratio of the number of likes to the number of followers and the ratio of the number of likes to the total number of posts of social robots are significantly higher than that of normal users. After analysing the information of social robot accounts, it is found that the suspicion that a user account belongs to a social robot account increases when the user account does not carry out authentication, the default profile is abnormal, the geolocation is abnormal, and the account description is abnormal.

Secondly, this paper explores the behavioural features of social robots and constructs a complete feature set, while the increase of features will enhance the complexity of the recognition model and lead to model overfitting. In order to improve the training speed of machine learning and eliminate the noise effect generated by irrelevant information, this paper chooses the feature selection method based on OOB estimation to eliminate redundant features, establishes a more reasonable and effective feature set, and improves the efficiency and stability of the model at the same time.

Finally, this paper uses the random forest algorithm, which has high accuracy, fast speed and stable performance, to construct a social robot recognition model, combining the four evaluation indexes of accuracy, precision, recall and F1 score to evaluate the model performance, and the experimental results show that the model constructed on the basis of the random forest algorithm has a higher accuracy and stability compared with other mainstream machine learning methods, such as decision trees and neural networks. The random forest model shows excellent performance in the experiments of social robot recognition, which can meet the requirements of the actual social robot recognition research and can be applied to the actual scenarios of robot account detection on social platforms.

## Author contributions

**Conceptualization:** Peng Zhang.

**Data curation:** Qilei Wang.

**Investigation:** Yinghui Du.

**Resources:** Ruiqing Qin.

**Writing – original draft:** Jiyang Zhang.

**Writing – review & editing:** Jiyang Zhang.

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
