## [Decision Letter · Decision Letter 0]

6 Dec 2024

PONE-D-24-40137Research on Social Bot Identification through Behavioral Feature AnalysisPLOS ONE

Dear Dr. Zhang,

Thank you for submitting your manuscript to PLOS ONE. After careful consideration, we feel that it has merit but does not fully meet PLOS ONE’s publication criteria as it currently stands. Therefore, we invite you to submit a revised version of the manuscript that addresses the points raised during the review process.

We look forward to receiving your revised manuscript.

Kind regards,

Sudipta Chowdhury

Academic Editor

PLOS ONE

Journal requirements: When submitting your revision, we need you to address these additional requirements. 1. Please ensure that your manuscript meets PLOS ONE's style requirements, including those for file naming. The PLOS ONE style templates can be found at https://journals.plos.org/plosone/s/file?id=wjVg/PLOSOne_formatting_sample_main_body.pdf and https://journals.plos.org/plosone/s/file?id=ba62/PLOSOne_formatting_sample_title_authors_affiliations.pdf 2. PLOS requires an ORCID iD for the corresponding author in Editorial Manager on papers submitted after December 6th, 2016. Please ensure that you have an ORCID iD and that it is validated in Editorial Manager. To do this, go to ‘Update my Information’ (in the upper left-hand corner of the main menu), and click on the Fetch/Validate link next to the ORCID field. This will take you to the ORCID site and allow you to create a new iD or authenticate a pre-existing iD in Editorial Manager. 3. Please note that PLOS ONE has specific guidelines on code sharing for submissions in which author-generated code underpins the findings in the manuscript. In these cases, we expect all author-generated code to be made available without restrictions upon publication of the work. Please review our guidelines at https://journals.plos.org/plosone/s/materials-and-software-sharing#loc-sharing-code and ensure that your code is shared in a way that follows best practice and facilitates reproducibility and reuse. 4. We note that the grant information you provided in the ‘Funding Information’ and ‘Financial Disclosure’ sections do not match.  When you resubmit, please ensure that you provide the correct grant numbers for the awards you received for your study in the ‘Funding Information’ section. 5. Thank you for stating the following financial disclosure:  [This work was supported by“Humanities and Social Sciences Foundation of the Ministry of Education (No. 22YJA860012)”, “Scientific Research Key Project of the Chinese People's Police University(No. ZDZX202201)”. We are grateful to the anonymous reviewers for their constructive comments that improving the quality of this work.].  Please state what role the funders took in the study.  If the funders had no role, please state: ""The funders had no role in study design, data collection and analysis, decision to publish, or preparation of the manuscript."" If this statement is not correct you must amend it as needed. Please include this amended Role of Funder statement in your cover letter; we will change the online submission form on your behalf. 6. Thank you for stating the following in the Acknowledgments Section of your manuscript: [This work was supported by“Humanities and Social Sciences Foundation of the Ministry of Education (No. 22YJA860012)”, “Scientific Research Key Project of the Chinese People's Police University(No. ZDZX202201)”. We are grateful to the anonymous reviewers for their constructive comments that improving the quality of this work.]We note that you have provided funding information that is not currently declared in your Funding Statement. However, funding information should not appear in the Acknowledgments section or other areas of your manuscript. We will only publish funding information present in the Funding Statement section of the online submission form. Please remove any funding-related text from the manuscript and let us know how you would like to update your Funding Statement. Currently, your Funding Statement reads as follows:  [This work was supported by“Humanities and Social Sciences Foundation of the Ministry of Education (No. 22YJA860012)”, “Scientific Research Key Project of the Chinese People's Police University(No. ZDZX202201)”. We are grateful to the anonymous reviewers for their constructive comments that improving the quality of this work.].  Please include your amended statements within your cover letter; we will change the online submission form on your behalf. 7. We note that your Data Availability Statement is currently as follows: [All relevant data are within the manuscript and its Supporting Information files.] Please confirm at this time whether or not your submission contains all raw data required to replicate the results of your study. Authors must share the “minimal data set” for their submission. PLOS defines the minimal data set to consist of the data required to replicate all study findings reported in the article, as well as related metadata and methods (https://journals.plos.org/plosone/s/data-availability#loc-minimal-data-set-definition). For example, authors should submit the following data: - The values behind the means, standard deviations and other measures reported;- The values used to build graphs;- The points extracted from images for analysis. Authors do not need to submit their entire data set if only a portion of the data was used in the reported study. If your submission does not contain these data, please either upload them as Supporting Information files or deposit them to a stable, public repository and provide us with the relevant URLs, DOIs, or accession numbers. For a list of recommended repositories, please see https://journals.plos.org/plosone/s/recommended-repositories. If there are ethical or legal restrictions on sharing a de-identified data set, please explain them in detail (e.g., data contain potentially sensitive information, data are owned by a third-party organization, etc.) and who has imposed them (e.g., an ethics committee). Please also provide contact information for a data access committee, ethics committee, or other institutional body to which data requests may be sent. If data are owned by a third party, please indicate how others may request data access.

Reviewers' comments:

Reviewer's Responses to Questions

**Comments to the Author**

1. Is the manuscript technically sound, and do the data support the conclusions?

Reviewer #1: Yes

Reviewer #2: Yes

2. Has the statistical analysis been performed appropriately and rigorously? 

Reviewer #1: Yes

Reviewer #2: Yes

3. Have the authors made all data underlying the findings in their manuscript fully available?

Reviewer #1: Yes

Reviewer #2: Yes

4. Is the manuscript presented in an intelligible fashion and written in standard English?

Reviewer #1: Yes

Reviewer #2: Yes

5. Review Comments to the Author

Reviewer #1: This research paper presents “ introduces a diverse array of behavioral features specific to social bots, drawing on the discernible disparities between their behaviors and those of human accounts. To refine the constructed feature set, an Out-of-Bag (OOB) estimation-based feature selection method is enlisted to eliminate redundant features. Concurrently, the study harnesses the Random Forest algorithm for its notable attributes of high accuracy, fast processing speed, and stable performance, thereby circumventing the inherent limitations of decision boundaries in standalone decision tree classifiers. Experimental findings highlight that the refined set of indicators sourced through the OOB estimation-based feature selection process contributes to enhancing the model’s overall stability. Notably, the Social Bot Identification model, built on the Random Forest framework, emerges as a superior alternative when contrasted with decision tree and neural network models in terms of accuracy and stability”

Good work keeps up

besides that, I have few minor comments which could further improve the quality of the manuscript

1. Provide quantitative remarks of the impact of the proposed method in the abstract.

2. need to rewrite clearly the contribution, motivation, challenges, your paper work.

3. write clear section for the literature review, and summarize it in table.

4. it is better to include a flow chart / pseudocode for your work.

5. The superiority performance of the proposed method could be achieved at what cost?

6.A detailed analysis of the limitations and potential failure scenarios of the proposed model is missing

7. Additional comparative analysis, around computational requirements and robustness of the model with other SOTA methods

Reviewer #2: The manuscript is robust and addresses a highly relevant topic; however, there are areas that could be improved to enhance its overall impact. The writing style could benefit from greater conciseness to reduce redundancy and ensure clarity, particularly in sections with dense technical descriptions. Simplifying these parts or providing additional context would make the study more accessible to readers outside the field. In addition, the introduction and literature review could be streamlined to focus more on directly relevant content and improve coherence.

Furthermore, the discussion of the results would be strengthened by elaborating on the practical implications and real-world applications of the proposed model. Ensuring that all figures and tables are paired with clear captions and well-integrated explanations within the text will also aid comprehension. Finally, a more focused conclusion that reiterates the key contributions of the study and suggests specific directions for future research would provide a stronger conclusion. These refinements will improve the clarity and attractiveness of the manuscript, bringing it into line with the standards of a high-impact journal.

6. PLOS authors have the option to publish the peer review history of their article (what does this mean? ). If published, this will include your full peer review and any attached files.

**Do you want your identity to be public for this peer review?** For information about this choice, including consent withdrawal, please see our Privacy Policy .

Reviewer #1: No

Reviewer #2: No

---

## [Author Response · Author response to Decision Letter 1]

21 Mar 2025

Dear reviewers,

Thank you for your valuable review comments on our paper.Your detailed suggestions have helped us to identify key parts of the article that need improvement and provided important guidelines for refining the paper.We have thoroughly revised and optimised the article, and the following is a point-by-point response to specific questions:

Reviewer 1:

Q1: Quantify the impact of the proposed method in the abstract.

A1: We have added a quantitative description in the abstract, which clearly lists the specific effects of the proposed method in improving model performance, so that readers can more intuitively understand the actual contribution of the method.

Q2: The contribution, motivation, and challenges of your dissertation work need to be rewritten clearly.

A2: We have completely rewritten the introduction section to present the research value of the paper more clearly.The motivation of the research work has been elaborated by combining the research background and practical problems in the current field, and the contribution points of the thesis as well as the technical challenges faced have been clarified at the same time.

Q3: Write a clear literature review section.

A3: We have made the literature review part independent of the second section, and have systematically sorted out and expanded the related studies to more clearly describe the progress of the existing studies and their limitations, so as to highlight the innovations of this paper's work.

Q4: It is better to attach a workflow diagram/pseudo-code.

A4: In order to improve the intuition of the paper, we have separated the methodology section into Section IV and added a detailed workflow diagram to show the specific steps of the proposed method.

Q5: At what cost is the superior performance of the proposed methodology achieved?

A5: We have supplemented the model analysis with relevant descriptions.By using the OOB method for feature filtering, the complexity of the features is significantly reduced, and despite a slight decrease in model performance, the overall level remains similar to that when using full features.Meanwhile, the computational complexity of Random Forest is comparable to other mainstream models while the performance is improved.

Q6: Detailed analyses of the limitations and potential failures of the proposed model are missing.

A6: Thanks for your correction.We have systematically analysed the limitations and potential failure scenarios of the model, such as feature dependency, limitation of the model's generalisation ability, and difficulties in the recognition of highly stealthy robots, in the new Section VI.At the same time, we present a detailed outlook on future improvement directions.

Q7: More comparative analysis with other SOTA methods around computational requirements and model robustness.

A7: The focus of this paper is to analyse the effectiveness of OOB feature selection methods, mainly focusing on the optimisation capability of feature engineering.The results of feature engineering are general in nature and are equally applicable to SOTA models.A better machine learning model (Random Forest) is chosen here just for the integrity of the testing system.

Reviewer 2:

We have made the following optimisation adjustments to the article based on your suggestions:

1. in order to improve the logical structure of the paper, the introduction and literature review are separated into two separate sections, which further clarify the research background and the shortcomings of the existing work, and highlight the innovations of this paper.

2. The methodology and experiments are separated into two sections to describe the proposed methodology and experimental design more systematically and to enhance the logical coherence of the content.

3. The experimental principles have been integrated into the methodology, which makes the content of the paper more compact and conforms to the writing standard of academic papers.

4. The overall language of the paper has been refined and embellished to enhance the clarity and professionalism of the presentation and to ensure that readers can clearly understand the core content and value of the study.

In addition, we have substantially revised the structure of the whole article to make it more in line with the writing structure of a research paper.Since we made changes to most of the article, we did not highlight the changes.

Finally, we would like to thank the reviewers again for their valuable comments on our paper.Your suggestions have greatly contributed to the quality of the paper and provided us with new ideas for our subsequent research.

Yours sincerely,

Peng Zhang

---

## [Decision Letter · Decision Letter 1]

28 Apr 2025

Research on Social Bot Identification through Behavioral Feature Analysis

PONE-D-24-40137R1

Dear Dr. Zhang,

We’re pleased to inform you that your manuscript has been judged scientifically suitable for publication and will be formally accepted for publication once it meets all outstanding technical requirements.

Kind regards,

Sudipta Chowdhury

Academic Editor

PLOS ONE

Additional Editor Comments (optional):

Reviewers' comments:

Reviewer's Responses to Questions

**Comments to the Author**

1. If the authors have adequately addressed your comments raised in a previous round of review and you feel that this manuscript is now acceptable for publication, you may indicate that here to bypass the “Comments to the Author” section, enter your conflict of interest statement in the “Confidential to Editor” section, and submit your "Accept" recommendation.

Reviewer #1: (No Response)

2. Is the manuscript technically sound, and do the data support the conclusions?

Reviewer #1: Yes

3. Has the statistical analysis been performed appropriately and rigorously? 

Reviewer #1: Yes

4. Have the authors made all data underlying the findings in their manuscript fully available?

Reviewer #1: Yes

5. Is the manuscript presented in an intelligible fashion and written in standard English?

Reviewer #1: Yes

6. Review Comments to the Author

Reviewer #1: good work keep up

all the comments are fulfilled but small comments must be addressed, which are submitted to the editor

7. PLOS authors have the option to publish the peer review history of their article (what does this mean? ). If published, this will include your full peer review and any attached files.

**Do you want your identity to be public for this peer review?** For information about this choice, including consent withdrawal, please see our Privacy Policy .

Reviewer #1: No

---

## [Editor Report · Acceptance letter]

PONE-D-24-40137R1

PLOS ONE

Dear Dr. Zhang,

I'm pleased to inform you that your manuscript has been deemed suitable for publication in PLOS ONE. Congratulations! Your manuscript is now being handed over to our production team.

Kind regards,

on behalf of

Dr. Sudipta Chowdhury

Academic Editor

PLOS ONE